# Neuromuscular and Biomechanical Jumping and Landing Deficits in Young Female Handball Players

**DOI:** 10.3390/biology12010134

**Published:** 2023-01-13

**Authors:** Maria Cadens, Antoni Planas-Anzano, Xavier Peirau-Terés, Ariadna Benet-Vigo, Azahara Fort-Vanmeerhaeghe

**Affiliations:** 1Research Group into Human Movement, National Institute of Physical Education of Catalonia (INEFC), University of Lleida (UdL), 25192 Lleida, Spain; 2Department of Sports Sciences, FCS Blanquerna, Ramon Llull University, 08022 Barcelona, Spain; 3Department of Sports Sciences, FPCEE and FCS Blanquerna, Ramon Llull University, 08022 Barcelona, Spain; 4Segle XXI Female Basketball Team, Catalan Federation of Basketball, 08950 Esplugues de Llobregat, Spain

**Keywords:** tuck jump, youth, asymmetry, girl, jump

## Abstract

**Simple Summary:**

Knowledge of neuromuscular and biomechanical deficits in young players is essential to improve the performance and reduce the risk of anterior cruciate ligament injury. The main aim of this study was to analyse and compare, by chronological age, jumping and landing deficits in young female handball players. A secondary aim was to relate the qualitative asymmetry to the quantitative asymmetry values detected. Sixty-one young female handball players performed the Tuck Jump Assessment test and the single leg countermovement jump. The youngest players presented greater neuromuscular and biomechanical deficits in the jumping and landing technique and also obtained highest asymmetry scores.

**Abstract:**

Neuromuscular and biomechanical imbalances that exist in jumping and landing actions should be examined in order to intervene to decrease the risk of ACL injury. The main aim of this study was to analyse and compare, by chronological age, jumping and landing deficits in young female handball players using the Tuck Jump Assessment (TJA). A secondary aim was to relate the qualitative asymmetry values detected using the TJA to the quantitative asymmetry values detected starting from the single leg countermovement jump (SL-CMJ). Sixty-one young female handball players (age: 14.3 ± 1.5 years) were distributed into three groups: U12, U14 and U16 and performed the TJA test and the single leg countermovement jump (SL-CMJ). The female U12 category players obtained the highest scores in the TJA and there were significant differences between the U12 (12.11 ± 1.97) and U14 (10.89 ± 1.74) categories (*p* = 0.017; ES = 0.374). In the U12 category, the female players presented larger interlimb asymmetry magnitudes in the SL-CMJ test; they also obtained higher scores in the qualitative criteria of the TJA test that referred to asymmetry (r = 0.43; *p* = 0.027). The analysis of the jumping and landing pattern using TJA allowed us to identify that the lower extremity valgus at landing, foot contact timing not equal and landing contact noise are the main biomechanical deficits in young female handball players. Furthermore, the asymmetry values assessed qualitatively (TJA) are associated with the asymmetry values assessed quantitatively (difference in jump achieved with each limb in the SL-CMJ test) in younger categories.

## 1. Introduction

Breakage of the anterior cruciate ligament (ACL) is one of the most serious injuries in view of its long-term consequences, due to the high risk of repeat injury [1], the long time required to return to competition and the degenerative changes and reduction in knee function [2]. The most common situations in which ACL injuries occur in handball are contactless actions involving jumping and landing or changes of direction [3]. Although the origin of injury is multifactorial, the first strategy to reduce risk must be to identify modifiable risk factors [4].

Deficits in muscle strength, power, coordination and muscle activation give rise to abnormal biomechanical movement patterns and to neuromuscular disorders which potentially increase the risk of ACL injury [4]. Hewett et al. [5], Myer et al. [6] and Fort-Vanmeerhaeghe et al. [7] classified them under six main categories: (1) leg dominance, which is dominance of the frontal compared with the sagittal plane as a dynamic control strategy of the lower limb in dynamic actions; (2) quadriceps dominance, which refers to dominance of muscle strength and activation of the extensor muscles compared with the flexor muscles of the knee [8]; (3) difference in muscle recruitment patterns, strength and coordination between the lower limbs, also called leg dominance; (4) trunk dominance, which is defined as the inability to control the position of the trunk in space in dynamic actions, which can lead to lateral movements of the trunk and thus increase knee abduction loads; (5) neuromuscular fatigue; and (6) reduction of the anticipatory or feedforward mechanisms.

These neuromuscular and biomechanical imbalances need to be examined and identified individually in female players in the early stages of growth, since this is when the ACL injury rate is highest [9]. The specific literature indicates anatomical, hormonal, biomechanical and neuromuscular control changes related to puberty as the main intrinsic risk factors explaining this higher incidence [10]. As they reach the last stages of pubertal maturation, their neuromuscular control deteriorates, giving rise to landings with less flexion of the knees and hips [8], leading to greater ground reaction forces [11]. They also present relative force deficits in the lower limbs, above all in the hamstrings [12] and lower activation of these muscles in relation to the quadriceps in landing actions, which potentially subject the ACL to a greater load in comparison with less mature girls [13]. 

An immature skeletal and muscular system, combined with the specific requirements of handball, can lead to specific unilateral adaptations of the sport, which can represent an intrinsic risk factor for ACL injury [14]. The asymmetry (leg dominance) between lower limbs has been a common line of research in recent years, covering three different aspects: (a) identify individuals who may be at risk of injury [15]; (b) monitor the function of the limbs during the rehabilitation process [16]; and (c) determine its impact on sport performance [17]. The types of assessment vary in complexity, from the simple height and distance of the jump to the three-dimensional analysis of the movement of the whole body, and, consequently, have a variety of cost and time implications [14]. The single leg countermovement jump (SL-CMJ) is one of those most frequently used in view of its validity and reliability [18]. Fort-Vanmeerhaeghe et al. [15] demonstrated that the interlimb differences in SL-CMJ could be a possible lower limb injury risk factor in team sports undertaken by young teens. On the contrary, the quantitative value of the asymmetry magnitude does not provide information on the quality of the movement. Assessing the landing technique of the lower limbs during a plyometric activity can provide information on neuromuscular and biomechanical deficits related to asymmetry [19]. The Tuck Jump Assessment (TJA) is an easy and cost-effective tool which allows errors to be identified in the landing technique of continuous jumps for 10′’ [19]. There are ten defects of technique which can be assessed qualitatively; three of them refer to asymmetry [6]. Assessing the level of motor competence during repeated jumping tasks, in which the female player is requested to respond to the disturbances and to the forces of the movement with presence of fatigue, may be more valid, from an ecological viewpoint, to identify ACL injury risk factors [20].

Fort-Vanmeerhaeghe et al. [7], Arundale et al. [21] and Benet-Vigo et al. [22] provide modified TJA values but consider the population by maturation age, sex or type of sport. Moreover, to date, no study has related the detection of asymmetries by grouping defects in the landing technique in TJA to the asymmetry magnitude assessed quantitatively using the SL-CMJ. Therefore, the main aim of this study was to analyse and compare, by chronological age, jumping and landing deficits in young female handball players using the TJA, and the secondary objective was to relate the qualitative asymmetry values detected using the TJA to the quantitative asymmetry values detected, starting with the SL-CMJ.

## 2. Materials and Methods

### 2.1. Participants

Sixty-one young female handball players participated voluntarily in this study (age: 14.3 ± 1.5 years; height: 1.59 ± 0.07 m; body mass: 54.3 ± 6.9 kg; years of PHV: 1.63 ± 1.09). The sample was classified according to their chronological age and the category in which they competed: U12 (n = 27; 44%), U14 (n = 27; 44%) and U16 (n = 7; 12%). Years of peak height velocity (PHV) were calculated by means of a regression equation which included age, body mass, standing height and sitting height measurements [23]. The maturational stage of each female player at the time of data collection was determined: early maturation (pre-PHV), defined as one year before PHV; average maturation (circa-PHV), ±1 year after PHV; and late maturation, over 1 year after PHV (post-PHV) [24]. The majority of the players were in a late maturation phase (49/61; 80%) (Table 1). 

All the participants underwent three training sessions per week, with an approximate duration of 120 min per session, and one match per week. The exclusion criteria were: (a) having previously had an ACL injury, (b) not having completed 80% of the training sessions during the month prior to the tests and (c) presenting any kind of injury (strain or acute) on the day of the tests.

All the participants and their respective legal guardians were informed of the procedures, methods, benefits and possible risks of participating in the study before giving their written consent (legal guardians). Furthermore, the research was undertaken with the consent of the managers of the handball clubs to which the female players belonged. This study was approved by the ethics committee of the Secretary General for Sport of Catalonia (20/2019/CEICEGC).

### 2.2. Procedure

The tests were carried out on the same day during the third month of the competition period (November). Previously, all participants had undertaken a familiarisation session to avoid any learning effect of the tests. Before beginning to warm up, the height, sitting height and body mass of each participant were recorded. The players then completed a 10 min warmup protocol based on the RAMP (raise, activate, mobilise and potentiate) system [25]. In the potentiate phase, 2 to 3 practice attempts of each test were performed. After the warmup, there was a three minute rest before recording data.

### 2.3. Measurements

#### 2.3.1. Single Leg Countermovement Jump

A contact platform (Chronojump, Bosco System, Barcelona, Spain) was used to calculate vertical jump capacity with one leg. Jump height was measured in metres starting from the flight time with a time resolution of 1 ms (1000 Hz) [26]. Each player was told to place themselves at the centre of the platform with the designated leg, with their hands on their hips and the other leg bent approximately 90° to avoid lateral propulsion [27]. When the player was ready, they had to undertake a countermovement at a self-selected depth before jumping as high as possible. They were also told to land with both feet simultaneously to increase the possibility of a stable landing. The attempt was not taken into account if they helped with the other leg, if the leg was not completely extended during the flight or if the hands were taken away from the hips [28].

The starting leg was randomised and three successful attempts were recorded for each leg and jump. There was a rest period of more than 60 s between each attempt. The best jump for each leg was used and the relative value was calculated in relation to the player’s body weight (SL-CMJr) for subsequent analysis [29].

#### 2.3.2. Tuck Jump Assessment

To analyse the jumping and landing technique, the test was recorded with two cameras (iPad 5 and iPhone 6s, Apple, Inc., Cupertino, CA, USA) at the height of the player’s hip. One camera was lined up three metres away on the sagittal plane and the other was lined up three metres away on the frontal plane. Before beginning the test, the players were shown a video and a live demonstration of the correct jumping technique. Each player was told to place their feet in a rectangle, 41 cm long by 35 cm wide, marked on the ground [7]. Each participant made continuous jumps for 10 s after receiving the basic instructions on how to perform the test. This included: (1) maximum and continuous jumps, (2) raising the knees above the height of the hips and (3) landing in the same rectangle with the feet shoulder width apart [19]. 

To allow a visible monitoring of the knees, the players had to wear shorts and sports shoes.

The TJA was scored by means of the modified version [6]. The scoring criteria and sheet are shown in Table 2. To rate an item, the alteration of the pattern had to occur on two or more occasions during the test, invalidating the first and last jump [6]. Items 1, 3, 4, 6 and 9 were assessed on the frontal plane, items 2 and 5 on the sagittal plane and items 7, 8 and 10 on both planes.

To facilitate understanding and subsequent analysis, in accordance with the aims of the study, the criteria assessed were grouped into categories of modifiable risk factors [31]. The category “leg dominance”, which refers to the concept of asymmetry, defined as an imbalance between the two lower limbs in regards to strength, coordination and motor control [6], comprised criteria (3) thighs not equal side to side during flight, (5) foot placement not parallel (front to back) and (6) foot contact timing not equal (asymmetric landing) (Figure 1).

Two raters (AM and MC), specialists in strength training with more than 10 years of experience and with prior experience in the assessment of the modified TJA, independently analysed and rated each of the criteria of the modified version of the TJA. They were allowed to play the videos as many times as required and at different speeds. If there was disagreement between the raters regarding the score allocation, a third rater with more experience (AF) decided on the final score.

### 2.4. Statistical Analysis

The mean and standard deviation (SD) were calculated for all variables. The assumptions of normality were checked using the Shapiro–Wilk test. The inter-rater measurement reliability for the scores of the 10 criteria of the TJA was analysed using the Kappa coefficient and was interpreted as: poor (≤0), mild (0.01–0.20), regular (0.21–0.40), moderate (0.41–0.60), good (0.61–0.80) and almost perfect (0.81–1) [32].

The total score of the TJA and the reliability of the intra-session measurements of the SL-CMJ test were analysed using the coefficient of variation (CV) and the intraclass correlation coefficient (ICC) type 2 with an absolute agreement and indicating 95% confidence intervals. The CV values are considered acceptable when CV ≤ 10% [33] and the interpretation of the values of the ICC were: ICC < 0.50 = poor, 0.50–0.74 = moderate, 0.75–0.90 = good and >0.90 = excellent [34]. 

The total scores of the TJ of each player were converted into a z-score, using the formula: z-score = (score of the participant in the TJ test—average score of the sample)/standard deviation. Reference points of the z-score of 0.2—0.99 were used (0.2 × standard deviation among participants) and ≥1 (1 × standard deviation among participants) to classify the results under scores “small to moderate” and “others and extreme”, respectively [35]. 

The associations for each of the 10 items of the TJA with the age of the participants were analysed by means of a chi-squared test and were interpreted as: irrellevant (<0.1), small (0.1–0.3), medium (0.31–0.5) and big (>0.5). To detect differences according to the age of the participants, the chi-squared test (χ^2^) and contingency tables were used. The Mann–Whitney test was moreover used to determine whether there were significant differences according to age in the overall score.

The magnitude of the interlimb asymmetry was quantified as the percentage difference between the two limbs according to the following equation: (100/(maximum value) × (minimum value) × − 1 + 100) [36]. 

The relationship between the asymmetry magnitude of the SL-CMJ test and the sum of the score for items 3, 5 and 6 (leg dominance) of the TJA test was assessed by means of an analysis of bivariate Pearson coefficients. 

The data were processed anonymously using a system of codes. The significance level was established as *p* < 0.05 for all the tests. The analyses were performed with the SPSS statistical program for Mac (version 27, IBM, New York, NY, USA) and JASP for Mac (version 14.1; JASP Team (2020), University of Amsterdam, Amsterdam, The Netherlands).

## 3. Results

The Kappa value according to the two raters in the TJA test was good to excellent for all items (Table 3). The ICC values for the total score of the TJA were excellent for all groups (between 0.92 and 0.95). The consistency between attempts for each age group in the SL-CMJ_L_ and SL-CMJ_R_ test was between 5.82% and 12.88%. All the ICC values were considered to be between good and excellent (between 0.82 and 0.96). Only the SL-CMJ_L_ test for the U16 category showed poor values (0.64) (Table 4). 

The female players from the U12 category obtained the highest scores in the TJA (12.11 ± 1.97) (Table 5) and significant differences were only observed between the U12 and U14 categories (*p* = 0.017; ES = 0.374). Three of the ten criteria of the TJA showed significant differences between categories, these being: lower extremity valgus at landing (criterion 1) (χ^2^ = 17.31; *p* = 0.002; V = 0.38), foot contact timing not equal (criterion 6) (χ^2^ = 10.47; *p* = 0.033; V = 0.22) and landing contact noise (criterion 7) (χ^2^ = 8.73; *p* = 0.013; V = 0.38) (Table 5). Eleven players had “high to extreme” values (z-score 13.41) and sixteen were “small to moderate” (z-score = 11.84) (Figure 2). 

The highest asymmetry magnitudes in the SL-CMJ test (14.88 ± 4.77) were observed in the U16 category (Table 6). Significant differences were not found between the asymmetry values of the SL-CMJ by category.

There was a directly proportional and statistically significant association in the U12 category between the asymmetry of the SL-CMJ test and the sum of items 3, 5 and 6 of the TJA test (r = 0.43; *p* = 0.027). The female players who obtained higher scores in the sum of the three criteria (3, 5 and 6) were also those who presented bigger interlimb asymmetry magnitudes in the unilateral vertical jump.

## 4. Discussion

The main aim of this study was to analyse and compare, according to chronological age, the jumping and landing patterns in young female handball players using the TJA. The main findings indicated that the players from the U12 category were those who presented higher values and, therefore, greater neuromuscular and biomechanical deficits in jumping and landing technique. The three criteria in which significant differences were observed between categories were: lower extremity valgus at landing (criterion 1), foot contact timing not equal (criterion 6) and landing contact noise (criterion 7). Second, the qualitative and quantitative asymmetry values detected were related using the TJA and the SL-CMJ, respectively. The female players from the U12 category who obtained higher scores in the sum of the three qualitative criteria of the TJA (qualitative value) which assessed asymmetry (3, 5 and 6) were also those who presented bigger interlimb asymmetry magnitudes in the SL-CMJ (quantitative value).

The mean of the overall score of the participants in the modified version of the TJA was 11.44 ± 1.97. These values were higher than in the studies by Benet-Vigo et al. [23] and Fort-Vanmeerhaeghe et al. [7] with girls of the same age; 8.82 ± 2.31 and 9.86 ± 2.01, respectively. This could be attributed to the nature of the sample of the present study, female handball players with less experience who did not undertake training sessions devoted exclusively to strength training. The highest total scores of the TJA were recorded in the U12 category, which was also the only category in which players of pre-PHV (n = 2/27) and circa-PHV (n = 10/27) maturation age were identified (Table 1). In the U14 and U16 categories, the mean of the total scores was lower than in the U12 category (Table 5) and all the players were already of post-PHV maturation ages. Taking into account the category in which the female players competed, significant differences were only found in the total value of the TJA in the U12 category in relation to the U14 category (*p* = 0.017; ES = 0.374). In the studies by Read et al. [37] and Fort-Vanmeerhaeghe et al. [7], the total score of the TJA went down as the chronological and maturation age of the participants went up. The improvement in the TJA score with age could be explained by neuromuscular and structural changes which occur during growth and maturation or by the fact that the players with more experience adapted positively to a greater training history [37].

In the original version of the TJA [19], the 10 criteria are assessed in a dichotomous manner and it is suggested that a total score above six should be the threshold to identify female players with a greater risk of injury and, therefore, the target of specific prevention training. In the present study, in which the modified version of the TJA was used [6], the participants with the highest scores in relation to the mean were identified by classifying them into two groups using the z-score calculation [23,36]. Eleven players out of the total obtained scores from “high to extreme” (z ≥ 1) and fifteen players obtained scores from “small to moderate” (z = 0.20 to 0.99) (Figure 2). The majority of the players with scores from “high to extreme” (8/11) and from “small to moderate” (8/15) were in the U12 category (Figure 2). The use and interpretation of the z-score allow identification of the players whose injury prevention strategies should be optimised [38] since, as demonstrated, the maturation phase and the chronological age of the female players can influence the jump landing performance [7].

The U12 category was the only one in which all participants performed the landing with accentuated valgus (the knees were touching) during the TJA (Table 6). In this same criterion (1), significant differences were found between the three categories (χ^2^ = 17.31; *p* = 0.002; V = 0.38). This neuromuscular deficit is attributed to a lower limb strength deficit which prevents the correct absorption of the forces generated in the jumping and landing actions and has a greater impact in girls and at early maturation ages [39,40]. Hewett et al. [39] indicate that the girls who present a greater valgus movement at landing have a high risk of ACL injury. These results confirm the need for neuromuscular training appropriate for the age in order to reduce the strength and motor control deficits with the aim of reducing female players’ injury risk [10,41].

Criteria 6 (foot contact timing not equal) and 7 (excessive landing contact noise) also showed significant differences between the three categories (χ^2^ = 10.47; *p* = 0.033; V = 0.22 and (χ^2^ = 8.73; *p* = 0.013; V = 0.38, respectively). Criterion 6 is grouped under the leg dominance neuromuscular risk factor [6], which is associated with lower limb asymmetries in the jumping and landing model. Over 70% of the sample in the U12 and U14 categories presented an asymmetric jumping arrival, in which one foot landed before the other. This neuromuscular deficit can be attributed to the nature of the sport, since its practice contributes to asymmetric development of the limbs [42]. In regards to criterion 7, no player was capable of performing the jumping arrivals without noise and only making contact with the tip of the foot, indicating that the study sample did not use sufficiently effective movement strategies to mitigate and dissipate landing forces [37]. 

The TJA contains 10 jumping and landing technique criteria rateable from 0 to 2. The criteria are added, obtaining an overall score. The bigger the resulting value, the more deficits the female player assessed will have in the jumping and landing model. Lininger et al. [43] and Arundale et al. [21] recommend not using the accumulated final value of the TJA as the only score, but rather identifying each risk factor described by each category of grouped criteria [6] (Figure 1). The biggest asymmetry score (3.71 ± 1.11) was detected in the U16 category, without there being significant differences between categories. In the case of the SL-CMJ, where the asymmetry was determined with a percentage value referring to the difference in jump between the two legs [44], the highest asymmetry values were also in the U16 category (14.88 ± 4.77). In the U16 category, despite having the highest asymmetry values in both the qualitative and the quantitative assessment, there was not a significant correlation between them; that is to say that the players who had greater asymmetry values in the SL-CMJ were not those who had higher asymmetry scores in the TJA. There was a significant correlation in the U12 category (r = 0.42; *p* = 0.027). The female players who presented high asymmetry values in the SL-CMJ also obtained a higher score in the asymmetry criterion of the TJA. These results, significant in early categories, suggest the need to include the qualitative assessment to identify which motor strategies the player uses to carry out the skill being developed, in this case jumping and landing. Populations who exhibit asymmetries between lower limbs have been associated with a higher incidence of injury [44], and ACL injuries during jumping tasks are more likely to occur during in the landing phase [13], so in training there should not only be the opportunity to improve the performance of the limb which shows a clear and consistent deficit compared to the other [42], but emphasis should also be placed on the technique; on how it is executed.

It should, however, be taken into account that the asymmetry is specific to the task [17,45,46], and therefore more than one test would be necessary to determine an asymmetry profile for a player and thus increase the sensitivity of the assessment. 

To date, there have not been any studies which correlate lower limb asymmetry assessed by means of a qualitative test, which gives information on how it is executed, and the difference in interlimb performance expressed as a percentage (magnitude). There are two studies conducted with a sample of similar characteristics (young girls) which related motor skills in jumping ability to performance. Sommerfield et al. [29] used the drop vertical jump (DVJ) as a tool to assess the landing model and found a negative association with the relative isometric force indicator (r = −0.35, *p* < 0.01); that is to say that the girls who presented a deficit in the landing model were those who had less relative isometric force. On the other hand, Pullen et al. [47] used the TJA as a qualitative tool to assess the jumping and landing model, but did not find any significant relation between motor skills (overhead squat, push-up, lunge, front support brace and shoulder Touch and Tuck Jump) and horizontal jump performance with both legs. 

Although this study compared a bilateral (TJA) with a unilateral (SL-CMJ) task, probably involving different motor strategies [48], the results indicate the need to prioritise training of the motor model in basic categories in order to create a good coordination strategy. The stabilisation of the landing to optimise muscle activation and thus ensure an adequate technique and alignment of the jumps (gentle landing with knees aligned) should be the first step before increasing the intensity and variability, for example, going from a bilateral to a unilateral task, involving different planes and axes, including disturbances with internal or external stimuli, combined with expected and unexpected actions to improve the feedforward capacity, increase the intensity of the muscle stretching and contraction cycle, combine elastic and reactive actions and progressively introduce the state of fatigue [49,50,51,52,53].

## 5. Limitations

There were three main limitations to the study. First, the sample was not classified according to maturation age. Second, for the U16 category, the sample was much smaller than for the U12 and U14 categories. The participants in the study belonged to teams organised according to their chronological age in which only two female players were identified in the pre-PHV phase in the U12 category of the competition, since girls reach PHV before boys [53]. Future studies should include equitable samples in the three maturation phases described by Sherar et al. [24]. Finally, only the relationship of the TJA and the SL-CMJ was investigated. Future studies should include more tests such as, for example, unilateral jumps with different directions to determine a more precise asymmetry profile in young female handball players. 

## 6. Conclusions

The results obtained from the analysis of the jumping and landing model present overall scores of the TJA of 11.44 ± 1.97, with significant differences between U12 and U14 categories. Criteria (1) dynamic valgus, (6) foot contact timing not equal and (7) landing noise were the most relevant and significant items between the three categories, and could be related to less relative force and motor control at younger chronological ages (U12) and during the early stages of maturation. The qualitative assessment of the TJA criteria grouped under the leg dominance or asymmetry risk factor is associated with the detection of quantitative asymmetries in the SL-CMJ test only in younger categories, notably U12.

## Figures and Tables

**Figure 1 biology-12-00134-f001:**
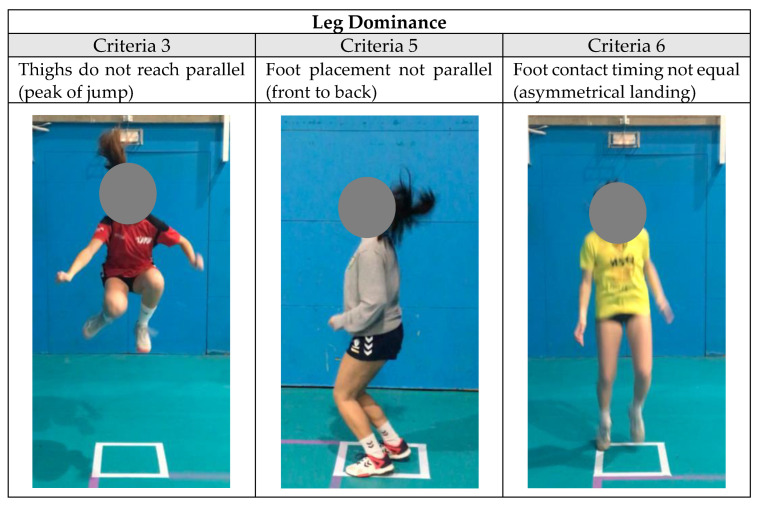
Tuck jump criteria grouped by risk factor category: leg dominance [6].

**Figure 2 biology-12-00134-f002:**
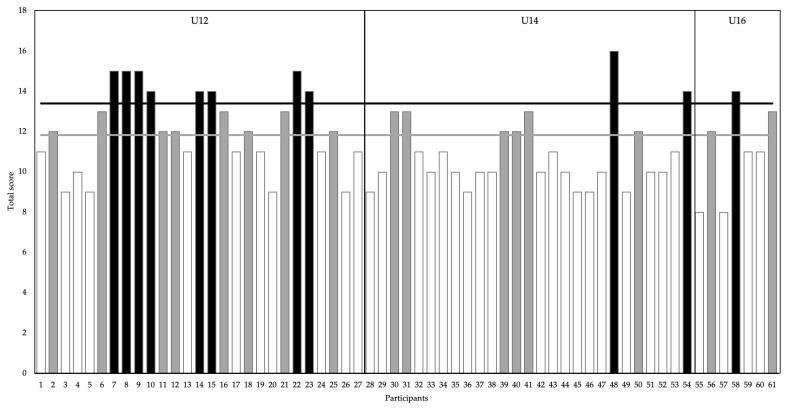
Overall score of the modified version of the TJA with z-score values. The grey line marks the threshold for “small to moderate” scores and the black line marks the threshold for “high to extreme” scores.

**Table 1 biology-12-00134-t001:** Descriptive data of the participants (mean ± standard deviation).

	Total (n = 61)	U12 (n = 27)	U14 (n = 27)	U16 (n = 7)
Chronological age (years)	14.3 ± 1.5	13 ± 1	15.1 ± 0.62	16.6 ± 0.3
Years of PHV ^1^	1.63 ± 1.09	0.79 ± 0.95	2.11 ± 0.48	3.01 ± 0.68
Pre-PHV ^2^	2 (3.23%)	2 (7%)		
Circa-PHV ^2^	10 (16.39%)	10 (37%)		
Post-PHV ^2^	49 (80.33%)	15 (56%)	27 (100%)	7 (100%)
Experience (years)	4.71 ± 2.50	4.14 ± 2.52	4.82 ± 2.51	6.43 ± 1.62
Body mass (kg)	54.34± 6.9	51.3 ± 10.7	55.9 ± 7.8	59.8 ± 9.1
Height (m)	1.59 ± 0.07	1.58 ± 0.07	1.60 ± 0.05	1.62 ± 0.11
BMI (kg/m^2^)	21.32 ± 3.12	20.56 ± 3.57	21.71 ± 2.67	22.72 ± 2.38

PHV: peak height velocity. ^1^ Estimation biological age [23]; ^2^ Number and percentage of participants.

**Table 2 biology-12-00134-t002:** Scoring criteria for each item of the Modified Tuck Jump Assessment [30].

Criterion	None (0)	Small (1)	Large (2)
(1) Lower extremity valgus at landing	No valgus	Slight valgus	Both knees touch
(2) Thighs do not reach parallel (peak of jump)	The knees are higher or at the same level as the hips	The middle of the knees are at a lower level than the middle of the hips	The whole knees are under the entire hips
(3) Thighs not equal side-to-side during flight	Thighs equal side to side	Thighs slightly unequal side-to-side	Thighs completely unequal side-to-side (one knee us over the other)
(4) Foot placement not shoulder width apart	Foot placement exactly shoulder width apart	Foot placement mostly shoulder width apart	Both feet fully together and touch at landing
(5) Foot placement not parallel (front to back)	Foot (the end of the feet) placement parallel	Foot placement mostly parallel	Foot placement obviously unparalleled (one foot is over half the distance of the other foot/leg)
(6) Foot contact timing not equal (asymmetrical landing)	Foot contact timing equal side-to-side	Foot contact timing slightly unequal	Foot contact timing completely unequal
(7) Excessive landing contact noise	Subtle noise at landing (landing on the balls of their feet)	Audible noise at landing (heels almost touch the ground at landing)	Loud and pronounced noise at landing (contact of the entire foot and heel on the ground between jumps)
(8) Pause between jumps	Reactive and reflex jumps	Small pause between jumps	Large pause between jumps (or double contact between jumps)
(9) Technique declines prior to ten seconds	No decline in technique	Technique declines after five seconds	Technique declines before five seconds
(10) Does not land in same foot print (consistent point of landing)	Lands in same footprint	Does not land in same footprint, but inside the shape	Lands outside the shape

**Table 3 biology-12-00134-t003:** Descriptive data of the participants (mean ± standard deviation).

Criterion	Kappa Value
1. Lower extremity valgus at landing	0.86
2. Thighs do not reach parallel	0.86
3. Thighs not equal side-to-side	0.75
4. Foot placement not shoulder width apart	0.77
5. Foot placement not parallel	0.72
6. Foot contact timing not equal	0.67
7. Excessive landing contact noise	0.64
8. Pause between jumps	0.83
9. Technique declines prior to 10 s	0.65
10. Does not land in same footprint	0.88

Note: Poor (≤0), mild (0.01–0.20), regular (0.21–0.40), moderate (0.41–0.60), good (0.61–0.80), almost perfect (0.81–0.99) and perfect [33].

**Table 4 biology-12-00134-t004:** Descriptive data of the participants (mean ± standard deviation).

	U12 (n = 27)		U14 (n = 27)		U16 (n = 7)	
	CV (%)	ICC (95% IC)	CV (%)	ICC (95% IC)	CV (%)	ICC (95% IC)
SL-CMJ_L_	12.88	0.82 (0.66–0.91)	8.22	0.96 (0.92–0.98)	10.58	0.64 (−0.01–0.93)
SL-CMJ_R_	10.7	0.89 (0.79–0.95)	7.48	0.96 (0.92–0.98)	5.82	0.95 (0.83–0.99)
TJA	4.78	0.92 (0.82–0.96)	5.34	0.92 (0.83–0.97)	4.61	0.95 (0.76–0.99)

SL-CMJ_L_: vertical countermovement jump with left leg; SL-CMJ_R_: vertical countermovement jump with right leg; TJA: Total score, Tuck Jump Assessment.

**Table 5 biology-12-00134-t005:** Number of participants who scored for each scoring criterion and the percentage (%). Descriptive values obtained, classified according to age.

C	U12 (n = 27)			U14 (n = 27)			U16 (n = 7)			*p*	χ^2^	V
	0	1	2	0	1	2	0	1	2			
1	**0**	**0**	27	0	8	19	1	1	5	0.002 *	17.31	0.38
	0%	0%	100%	0%	30%	70%	14%	14%	71%			
2	5	8	14	9	11	7	0	3	4	0.172	6.39	0.23
	19%	30%	52%	33%	41%	26%	0%	43%	57%			
3	0	20	7	0	16	11	0	4	3	0.458	1.56	0.16
	0%	74%	26%	0%	59%	41%	0%	57%	43%			
4	3	16	8	2	19	6	0	7	0	0.358	4.37	0.19
	11%	59%	30%	7%	70%	22%	0%	100%	0%			
5	0	19	8	0	24	3	0	5	2	0.222	3.01	0.22
	0%	70%	30%	0%	89%	11%	0%	71%	29%			
6	6	20	1	8	19	0	2	3	2	0.033 *	10.47	0.29
	22%	74%	4%	30%	70%	0%	29%	43%	29%			
7	0	10	17	0	19	8	0	6	1	0.013 *	8.73	0.38
	0%	37%	63%	0%	70%	30%	0%	86%	14%			
8	16	11	0	17	10	0	7	0	0	0.120	4.23	0.26
	59%	41%	0%	63%	37%	0%	100%	0%	0%			
9	0	24	3	0	23	4	1	5	1	0.089	8.06	0.26
	0%	89%	11%	0%	85%	15%	14%	71%	14%			
10	0	25	2	0	25	2	0	7	0	0.758	0.56	0.10
	0%	93%	7%	0%	93%	7%	0%	100%	0%			

C: Criterion; V: Cramer’s V, irrellevant (<0.1), small (0.1–0.3), medium (0.31–0.5) and big (>0.5). * Significant differences by ages.

**Table 6 biology-12-00134-t006:** Mean values for performance and for the asymmetry risk factor of the TJA and SL-CMJ.

	Total (n = 61)	U12 (n = 27)	U14 (n = 27)	U16 (n = 7)
SL-CMJ_L_	13.85 ± 2.92	13.52 ± 2.53	14.07 ± 3.58	14.29 ± 1.04
SL-CMJ_R_	13.36 ± 2.92	13.48 ± 2.69	13.36 ± 3.33	12.94 ± 2.37
ASIM SL-CMJ (%)	12.05 ± 8.29	11.55 ± 9.29	11.80 ± 8.05	14.88 ± 4.77
TJA	11.44 ± 1.97	12.11 ± 1.97	10.89 ± 1.74	11 ± 2.31
ASIM TJA	3.34 ± 0.89	3.37 ± 0.93	3.22 ± 0.80	3.71 ± 1.11

SL-CMJ_L_: vertical countermovement jump with left leg; SL-CMJ_R_: vertical countermovement jump with right leg; ASIM SL-CMJ(%): asymmetry magnitude in the vertical single leg countermovement jump test expressed with percentage; TJA: Total score, Tuck Jump Assessment; ASIM TJA: sum of criteria 3, 5 and 6 corresponding to the asymmetry risk factor (leg dominance).

## Data Availability

Data are available on request due to restrictions, e.g., privacy or ethical restrictions.

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
