# Peer review of "Neuromuscular and Biomechanical Jumping and Landing Deficits in Young Female Handball Players"

_biology, 2023, doi:10.3390/biology12010134_

Round 1

Reviewer 1 Report

Congratulations for the article. 

Although the research topic is not a novelty, the analysis that the authors present, regarding the detection of asymmetries by grouping defects in the landing technique of handball players in TJA and relating them to the asymmetry magnitude assessed quantitatively using the SL-CMJ is quite interesting. Also, the authors made a great job using the statistical analysis and added an inter-observer study to support their results. Congratulations.

Reviewer 2 Report

First of all, I would like to congratulate the authors for their efforts.

-As you know, there are many studies conducted with derivatives of this research.

-Asymmetry, etc. from CMJ-SJ. A lot of work was done using many parameters produced. It is not clear what is your biggest innovation.

-To be honest, things sound good, but there is something missing. The method is complex and there is no flow unfortunately.

-Especially if you revise the innovations and method, I would like to examine it once again.

-The quality of tables and graphics should be improved. 

I want to review it once more.

Reviewer 3 Report

The manuscript entitled "Neuromuscular and biomechanical jumping and landing deficits in young female handball players" discusses an interesting and actual topic, that is the assessment of asymmetries in sports, and its relation with age and sex differences. Indeed, this allows trainers to adapt their training and physiotherapists to adapt their treatment with a more tailored approach.

In addition, this work describes the characteristics of a sport that is not very widely represented in research.

I have some comments/suggestions, I hope the authors might find helpful to improve the manuscript:

Abstract:

- If there is some space (word count), I suggest adding a sentence on the background that justifies the aims, and not only stating the aims of the study without a rationale.

- Age does not require two decimals; the authors can leave just one decimal.

- Although worth to be mentioned, the "criteria" have not been described in the methods of the abstract; in particular, I find quite confounding the numbering they gave (...3) thighs not equal...5) foot placement...or...valgus at landing (criterion 1)...). Just describe what you found, without indicating any number since it is not described elsewhere in the abstract text.

Introduction:

- line 63: the specialized literature...I do not think that "specialized" should be used, as scientific literature should be always "specific", otherwise it should not be published.

- line 88: I think that the whole paragraph about the description of qualitative evaluation at the TJA should better fit in the methods, rather than in the introduction.

Methods:

- as before, I suggest having only 1 decimal for age and body mass; body height should be reported in meters as for the ISU; first define PHV

- since the age of the participants, was the menstrual cycle taken into consideration? were all the participants tested during the same phase?

- Can the authors report additional details about the platform, such as the sampling frequency? They can be useful for better reproducibility of the findings.

- Table 2, I suggest redrawing it as it seems that in the uploaded version some rows are too close to each other and are poorly readable.

- I want to congratulate the authors on the careful statistical analysis performed and the well-described section

Results:

- Line 226, maybe SL-CMJD is SL-CMJR?

- Figure 2 looks somehow confusing; in addition, the x-axis label is titled "Title"; what about drawing a boxplot/scatterplot instead?

Discussion:

- The authors describe the importance of such findings and assessment in terms of findings on subjects with a higher risk for injuries; while for the TJA analysis this has been discussed, the discussion could describe some of the previous findings reporting the hypothesis that asymmetries could be associated with increased risk of injuries in team sports (Alentorn-Geli et al., Knee Surg Sports Traumatol Arthrosc, 2015; Buoite Stella et al., Sports, 2022; Owoye et al., Sports Med Open, 2020).

Round 2

Reviewer 2 Report

I think the authors enhanced the research. I think it can be published.

Reviewer 3 Report

I am fine with the responses and edit made. Thank you.